# Enhancing Multi-Domain Recommendations via LLM-Generated Data

**Chumeng Jiang 2024316092**   **Kairong Luo 2024310643**   **Zhixuan Pan 2024311550**

## 1 Introduction

Due to the increasingly severe information overload issue, the application of recommendation systems (RS) prevails across all kinds of Internet platforms, as they can provide personalized items for each user. Recently, with the growing number of specialized domains in one comprehensive platform, such as short video recommendations, article recommendations, and product recommendations in the same app, multi-domain recommendation has garnered significant attention. The multi-domain recommendation can simultaneously leverage knowledge from different domains, alleviating the data sparsity issue and allowing a single model to make recommendations across multiple domains, reducing the deploying costs.

Nevertheless, the historical data size between different domains varies. Some areas may have significantly more data than others, namely the rich or cold-start scenarios. This disparity in data size can lead to certain limitations during model training. For instance, the learning of domain-specific parameters in cold-start scenarios may be insufficient, while the learning of domain-shared parameters may be dominated by rich scenarios. Previous work mainly addresses these issues through meticulous structural design of the models [5, 1, 13, 17].

In this paper, we adopt a different perspective by addressing this issue from the data standpoint. The emergence of LLMs has made it possible to generate virtual user and item data. Furthermore, LLMs, with their extensive world knowledge and outstanding comprehending capability, have demonstrated impressive recommendation capabilities in cold-start scenarios [3]. In this case, we utilize the LLMs to simulate users in cold-start scenarios and synthesize more sufficient positive samples after learning from existing multi-domain historical interactions. Through an elaborately designed data filtering and denoising strategy, the recommendation quality of the multi-domain models can be enhanced. Moreover, through the lens of recommendation systems, we may get more insight into the synthetic data from existing LLMs [12, 10].

## 2 Related Work

### 2.1 Multi-Domain Recommendation

Extensive work has been done to address the challenge of recommendation within multi-domain scenarios. The multi-domain recommendation is a cross-domain recommendation approach aimed at enhancing accuracy across multiple domains simultaneously. For instance, Domain generalization (DG) methods extract common knowledge from various domains, thereby improving generalization to unknown domains and mitigating data sparsity issues [17]. STAR [13] enhances the capture of the features of each domain by introducing Partitioned Normalization (PN) and domain-specific fully connected networks (FCNs) to capture the unique characteristics of each domain.

However, many of these methods struggle with effectively disentangling domain-shared and domain-specific knowledge. Methods like CATART [5] utilize auto-encoders to create global embeddings from domain-specific ones, relying on attention mechanisms to aggregate these embeddings for recommendations. However, this approach can inadvertently compromise disentanglement, as domain-shared knowledge influences the updating of domain-specific embeddings. Similarly, SAML [1]

Preprint. Under review.

maps features into global and domain-specific embeddings and employs a mutual unit to learn domain similarity, yet it lacks effective alignment mechanisms and fails to fully exploit the inter-domain relationships. Overall, while various methodologies exist, challenges related to model complexity and data sparsity persist.

## 2.2 Data Synthesis in Recommendation

Many traditional synthetic data generation methods have been proposed to address data imbalance in RS, i.e. the issues of the data sparsity and the long-tail distribution in the data. Common methods [14] include using k-nearest neighbors to create new instances based on existing minority class samples, employing generative models like GANs, VAEs, and diffusion models to generate synthetic tabular data. The data synthesized by these traditional methods often closely resembles the distribution of the original dataset.

Using LLMs to synthesize data allows for the incorporation of LLMs' inherent world knowledge and the utilization of more textual information during the synthesis process. Synthetic data generated by LLMs has been applied in fields such as healthcare [7, 16], demonstrating certain effectiveness. In the recommendation field, attempts to utilize data synthesized by LLMs are still relatively limited. ONCE [9] prompts closed-source LLMs to synthesize the items that new users with fewer than five historical records might be interested in, enriching their data to achieve better user embeddings. LLMRec [15] conducts the data augmentation by requiring the LLMs to select the most likely positive and negative item pairs within the candidate set provided. It leverages LLMs' advantages in pointwise comparison and employs MAE and noise pruning for denoising. However, there is no previous work that utilizes LLM-generated data to promote multi-domain recommendation. There are challenges including balancing the data density across multiple domains, controlling the noises introduced from synthetic data, and aligning the LLMs with the multi-modal data distributions in the multi-domain RS scenarios.

## 3 Problem Formulation

We target one of the most common problems in multi-domain recommendation: CTR prediction. The recommender system uses interaction data $(d, \mathbf{x})$ to predict $y$ under the multi-domain setting, where $y$ indicates whether a click occurred ($y = 1$) or not ($y = 0$). Here, $d$ is the domain indicator, distinguishing samples from $D$ total domains, and $\mathbf{x}$ consists of raw features, including user and item attributes. Assuming there are $M$ categorical features, $\mathbf{x} = [\mathbf{x}_1, \mathbf{x}_2, \ldots, \mathbf{x}_M]$, with $\mathbf{x}_m$ being the one-hot representation of the $m$-th feature. An embedding layer maps $\mathbf{x}$ to a low-dimensional vector $\mathbf{e} = [\mathbf{e}_1 \parallel \mathbf{e}_2 \parallel \ldots \parallel \mathbf{e}_M]$, where $\parallel$ indicates concatenation. For the $m$-th feature, $\mathbf{e}_m$ is derived through a learnable lookup operation $\mathbf{e}_m = \mathbf{E}_m \cdot \mathbf{x}_m$, with $\mathbf{E}_m \in \mathbb{R}^{k \times \mu_m}$ as the weight matrix, $k$ as the embedding size and $u_m$ as the number of feature values. Finally, the predicted result $\hat{y}$ for whether a user will click on an item is computed as $\hat{y} = f_d(\mathbf{e})$, where $f_d$ is the recommendation model for the $d$-th domain. We use the Negative Log-Likelihood Loss, which is also known as the binary cross-entropy loss $\mathcal{L}(\boldsymbol{\Theta}) = -\frac{1}{B} \sum_{i=1}^{B} [y_i \log(\hat{y}_i) + (1 - y_i) \log(1 - \hat{y}_i)]$, where $\Theta$ represents the learnable parameters, $B$ is the batch size, and $y_i$ and $\hat{y}_i$ are the true label and predicted result for the $i$-th sample, respectively.

## 4 Method

- **Datasets:** The *Amazon Datasets*[1] consist of product and user metadata, and users' product reviews from the Amazon platform. There are 29 categories in total, with some overlapping users across different categories. Recommendations for products in each distinct category can be viewed as a separate domain.

- **Recommendation Models:** (1) Single-domain models: SASRec [4] and LightGCN [2]; (2) Multi-domain models: EMCDR [11], BiTCDR [8] and CUT [6].

- **Evaluation Metrics:** We choose *AUC* as the measure of recommendation accuracy, while implementing the *Gini coefficient* and *item coverage* to examine the user-side and item-side recommendation fairness, respectively.

---

[1]https://nijianmo.github.io/amazon/index.html

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
