# OpenReview forum: "[Proposal] Enhancing Multi-Domain Recommendations via LLM-Generated Data"
_tsinghua.edu.cn/THU/2024/Fall/AML — THU 2024 Fall AML Submission_

### Official Review · ~Ziang_Zheng1 · 2024-11-06
**Leveraging LLMs for Data Synthesis in Multi-Domain Recommendation: A Promising Data-Centric Approach with Considerations for Quality and Fairness**

**Rating:** 8
**Confidence:** 3

**Review:**

**Summary**
This paper addresses the data sparsity issue in multi-domain recommendation systems (RS) by proposing an approach that leverages large language models (LLMs) to generate synthetic user and item data in cold-start scenarios. By focusing on data synthesis rather than structural model adjustments, the authors aim to improve recommendation quality and reduce deployment costs across multiple domains.

**Strengths**
1. **Novelty**: The use of LLMs for generating virtual user and item data to mitigate cold-start issues in multi-domain RS is a novel approach. The authors bring a unique perspective by focusing on the data itself rather than only refining model architectures.
2. **Timeliness**: Multi-domain recommendation is increasingly important due to the rise of comprehensive platforms offering various services. This work is timely and addresses a real-world problem.
3. **Solid Motivation**: The authors clearly explain the challenges of data imbalance across domains and how LLMs, with their extensive knowledge base, offer a feasible solution for generating high-quality data.
4. **Comprehensive Related Work**: The related work section is thorough and demonstrates a strong grasp of prior research in both multi-domain RS and synthetic data generation.

**Weaknesses**
1. **Data Quality and Filtering**: The proposal briefly mentions filtering and denoising synthetic data generated by LLMs. However, it lacks a detailed methodology on how the authors plan to address potential noise and bias in the generated data, which is crucial for ensuring RS performance.
2. **Potential LLM Biases**: LLMs may introduce unintended biases or inaccuracies in generated data, especially for nuanced user preferences in multi-domain scenarios. The proposal would benefit from a discussion on mitigating such biases.
3. **Evaluation Plan**: While the proposal includes AUC, Gini coefficient, and item coverage metrics, it lacks clarity on how these metrics will capture the impact of synthetic data on recommendation fairness across domains.

**Overall Evaluation**
The proposal presents a promising approach to improving multi-domain recommendation systems using LLMs for data synthesis. With additional clarity on data filtering and bias mitigation, the proposed methodology has strong potential to advance multi-domain RS research.

**Score**
- **Originality**: 8/10
- **Clarity**: 7/10
- **Technical Soundness**: 6/10
- **Relevance**: 9/10

**Recommendation**: Accept with minor revisions.

---

### Official Review · ~Huajun_Bai1 · 2024-11-07
**Advancing Multi-Domain Recommendations with LLM-Synthesized Data: A Review of Innovation and Its Challenges**

**Rating:** 5
**Confidence:** 3

**Review:**

Strengths
1. Novelty: The paper presents an innovative approach to multi-domain recommendation systems by leveraging LLMs to generate synthetic data, addressing the cold-start problem in a unique way.
2. Comprehensive Literature Review: The authors have provided an extensive review of related works, mentioning numerous papers that cover multi-domain recommendation and data synthesis, demonstrating a solid understanding of the field.
3. Practical Relevance: The paper tackles the real-world issue of data sparsity in multi-domain RS, which is particularly pertinent given the increasing number of platforms offering a variety of services.

Weaknesses
1. Data Filtering and Denoising: The proposal lacks a detailed explanation of the methodology for filtering and denoising the synthetic data generated by LLMs, which is essential for ensuring the quality and reliability of the recommendation system.
2. Multi-Domain Evaluation Metrics: Although the paper employs established metrics such as AUC, Gini coefficient, and item coverage, there is a need for a more explicit discussion on how these metrics will specifically assess the impact of synthetic data on the fairness and accuracy of recommendations across multiple domains.

---

### Official Review · ~Juncheng_Yu1 · 2024-11-07
**Advancing Multi-Domain Recommendation through Synthetic Data Generation with LLMs: A Promising Data-Driven Solution**

**Rating:** 7
**Confidence:** 3

**Review:**

## Summary

This paper addresses the challenge of unbalanced data across different domains in multi-domain recommendation systems by generating synthetic data using large language models. With carefully designed data filtering and denoising strategies, this approach is believed to have the potential to significantly enhance multi-domain recommendation performance.

## Strengths

- **Relevance of Research Problem**: In introduction, the authors show a strong understanding of the recommendation domain, supported by a thorough literature review. They have clearly identified a precise and significant problem within the field.

- **Clarity of Task Definition**: In methods and problem formulation, the authors have clearly illustrated the task, providing a detailed pipeline to solve the problem, effectively formulating the research objective.

## Weaknesses

- **Ambiguity when mentioning CTR Prediction**: In the first sequence of problem formulation, It is unclear if the formulation that follows the initial problem statement pertains directly to CTR prediction. Clarification would enhance the coherence of this section.

- **Lack of Clarity in Key Methodological Innovations**: Although the Related Work section extensively covers prior research on LLM-enhanced recommendation, the Problem Formulation and Method sections lack clarity on how LLMs **in their work** are applied to enhance this specific problem. The authors introduce the problem, datasets, model, and metrics of the recommendation, but omit a clear explanation of how the LLM generates data and how this process enhances the recommendation system.

- **Potential for Model Pattern Disruption from Generated Data**: Recent researches on utilizing generated data to train large language figure out that the use of synthetic data from LLMs may risk altering the model’s inherent patterns. It would be better if they add additional discussion or experimentation to demonstrate mitigation of this risk, which can also strengthen the contribution.

## Score

- **Soundness**: 8/10

- **Contribution**: 7/10

- **Presentation**: 6/10

---

### Official Review · ~Kangping_Xu1 · 2024-11-09
**Review of "Enhancing Multi-Domain Recommendations via LLM-Generated Data"**

**Rating:** 8
**Confidence:** 4

**Review:**

## Pros

1. **Novel Approach to Data Imbalance**
   - Takes a fresh perspective by addressing multi-domain recommendation challenges through data enhancement rather than model architecture
   - Leverages the emerging capabilities of LLMs in a practical application, potentially creating a new direction for recommendation system research

2. **Comprehensive Evaluation Framework**
   - Proposes a well-rounded evaluation strategy using both accuracy (AUC) and fairness metrics (Gini coefficient, item coverage)
   - Includes comparison with both single-domain and multi-domain baseline models, providing a thorough validation approach

## Cons

1. **Limited Technical Details**
   - The method part lacks specific details about the LLM data generation process and the "elaborately designed data filtering and denoising strategy" mentioned in the introduction, which I think is the most important enhancement and
   - There is no clear explanation of how the synthetic data will be validated for quality and authenticity. I am unsure if just evaluating the end-to-end recommendation metrics would guarantee the quality of the synthetic data.

The proposal presents an interesting direction for research but would benefit from more detailed technical specifications and consideration of practical implementation challenges.

---

### Official Review · ~Guangjie_Xu1 · 2024-11-09

**Rating:** 8
**Confidence:** 4

**Review:**

**Summary**

Overall, this proposal is robust, addresses a critical research gap, and suggests a promising approach to an emerging issue in recommendation systems.

**Research Problem**

The problem is both innovative and relevant, exploring LLMs’ applications beyond traditional NLP, positioning the research at the frontier of recommendation system technology. This approach taps into a high-interest area, capitalizing on recent trends in leveraging LLMs to address data limitations.

**Reliability of the Idea**

The proposal’s concept appears reliable. Recent studies show LLMs can simulate user behavior effectively with minimal historical data, making the application of LLM-generated data for cold-start scenarios a well-founded choice.

**Plan**

The proposal outlines specific models and evaluation metrics, providing a clear roadmap for implementation. However, detailing how the data filtering strategy will handle differences across domains would further strengthen the plan.

**Writing**

The language is generally precise and professionally written, though some technical terms could benefit from further clarification for readability. Most sections are well-explained and demonstrate a clear, logical flow.

---

### Official Review · ~Liutao7 · 2024-11-09
**The proposal has a good concept, exploring the use of large models to solve some practical problems of recommendation systems from a data perspective.**

**Rating:** 9
**Confidence:** 4

**Review:**

The proposal has good application potential and presents an interesting idea of using large language models to generate data to enhance the performance of multi-domain recommendation systems. I think: 1) The proposal has good integrity. 2) The proposal explores the potential of LLM in generating virtual user and item data, which offers a novel approach to addressing the issue of data sparsity. 3) The workload is reasonable.

One suggestion: It would be important to further refine the dataset construction method in the proposal; also, consider whether the issue of hallucinations in large models itself needs to be addressed, and how to verify the quality and authenticity of the generated data.

---

### Official Review · ~Tong_Yu9 · 2024-11-10
**Review of "Enhancing Multi-Domain Recommendations via LLM-Generated Data"**

**Rating:** 8
**Confidence:** 3

**Review:**

Quality: The quality of the paper is generally high, the methodology is clear, and the effective solution is proposed for the sparse data problem in the multi-domain recommendation system. However, the details of the data generation and implementation process are not well described.

Originality: Although the idea of using LLM to generate data was novel, its distinction and advantage from existing methods needed to be further emphasized in the application of multi-field recommendation system.

Significance: This research is of great significance in multi-domain recommendation systems, especially in cold start scenarios. However, the representativeness of the generated data may affect its practical application.

Pros:
Innovative approach: The idea of using LLM to generate virtual user and project data is forward-looking.
Solve the problem of data sparsity: This method effectively alleviates the problem of data sparsity in multi-domain recommendation.

Cons:
Lack of originality: Although the method is novel, it lacks obvious innovation points compared to existing research.
Complexity: The complexity of data generation and denoising can affect the feasibility of practical applications.
Data representation problem: The generated data may not fully represent the real user behavior, affecting the recommendation effect.

---

### Official Review · ~Changsong_Lei2 · 2024-11-11
**Review of "Enhancing Multi-Domain Recommendations via LLM-Generated Data"**

**Rating:** 8
**Confidence:** 4

**Review:**

### Summary:
This proposal tries to address the challenge of data sparsity and imbalance in multi-domain recommendation systems by leveraging synthetic data generated by large language models (LLMs).

### Pros:
- The proposal demonstrates a strong understanding of the technical requirements for multi-domain recommendations. It outlines methods for data synthesis, noise control, and the incorporation of LLM-generated data within the system.
- The CTR prediction task is clearly defined, and the paper provides necessary mathematical notations to describe the problem and evaluation metrics, such as AUC, Gini coefficient, and item coverage.

### Cons:
- Given the computational intensity of LLMs, it would be useful to discuss the scalability of the approach..
- Lack clear description for their proposed method and motivation.

Generally speaking, this proposal is well written and demonstrates their understanding of the problem.

---

### Official Review · ~Jin_Zhu_Xu1 · 2024-11-11
**Clear Motivation and Well Define Problem**

**Rating:** 8
**Confidence:** 4

**Review:**

The proposal points out a clear problem and motivation for the idea however, the methodology is not convincing enough. It would be better if the proposal thoroughly included more concrete details of techniques implementation and the roles of LLM. Although the author tried to explain briefly how synthetic data can improve the recommendation quality of the current models, the explanation of technical terms is somewhat unclear and there is a notable lack of detail in the implementation approach.

---

### Official Review · ~Yuji_Wang4 · 2024-11-12
**Review of "Enhancing Multi-Domain Recommendations via LLM-Generated Data"**

**Rating:** 8
**Confidence:** 3

**Review:**

The project aims to explore the research topic of multi-domain recommendations. The authors propose to address the problem raised by imbalanced data size between different domains by enhancing the training with LLM-generated data.

### Strengths:

1. Topic selection: Multi-domain recommendation is a research topic with high practical value. The proposal highlights the challenges in this area and offers targeted solutions that address these difficulties.
2. Well-structured writing: The proposal is well-organized. It provides a comprehensive review of related works and the research problem is well-defined. Meanwhile, the experimental plan is detailed and feasible, providing a solid foundation for the study.

### Weaknesses:

1. Using LLM-generated data to enhance the training process is a widely-used approach. It could be better if innovative methods can be designed when implement this approach.
2. The method section lacks specific details about the LLM used for data generation and the algorithms for data augmentation and preprocessing.

---

### Official Review · ~Chendong_Xiang1 · 2024-11-12

**Rating:** 7
**Confidence:** 2

**Review:**

Strengths:
	1.	Innovative Use of LLMs for Data Generation: The paper takes a fresh approach by using LLMs to generate synthetic data, tackling the data sparsity issue in multi-domain recommendations.
	2.	Effective Cold-Start Handling: It addresses the cold-start problem by creating simulated user interactions, improving recommendation quality in low-data domains.
	3.	Comprehensive Evaluation: The study uses multiple metrics (AUC, Gini coefficient, item coverage) to assess accuracy and fairness, providing a well-rounded evaluation.
	4.	Cost Reduction Potential: A single, cross-domain model may reduce deployment and maintenance costs, benefiting platforms with diverse domains.

Weaknesses:
	1.	Limited Domain-Specific Analysis: The impact of synthetic data on different domains isn’t thoroughly explored; some domains may benefit more than others.
	2.	Noise from Synthetic Data: The risk of noisy data impacting performance remains, and the denoising process could be explained more clearly.